# Enhanced Clearance of Neurotoxic Misfolded Proteins by the Natural Compound Berberine and Its Derivatives

**DOI:** 10.3390/ijms21103443

**Published:** 2020-05-13

**Authors:** Paola Rusmini, Riccardo Cristofani, Barbara Tedesco, Veronica Ferrari, Elio Messi, Margherita Piccolella, Elena Casarotto, Marta Chierichetti, Maria Elena Cicardi, Mariarita Galbiati, Cristina Geroni, Paolo Lombardi, Valeria Crippa, Angelo Poletti

**Affiliations:** 1Dipartimento di Scienze Farmacologiche e Biomolecolari (DiSFeB), Dipartimento di Eccellenza 2018-2022, Centro di Eccellenza sulle Malattie Neurodegenerative, Università degli Studi di Milano, 20133 Milan, Italy; paola.rusmini@unimi.it (P.R.); riccardo.cristofani@unimi.it (R.C.); barbara.tedesco@unimi.it (B.T.); veronica.ferrari@unimi.it (V.F.); elio.messi@unimi.it (E.M.); margherita.piccolella@unimi.it (M.P.); elena.casarotto@unimi.it (E.C.); marta.chierichetti@unimi.it (M.C.); MariaElena.Cicardi@jefferson.edu (M.E.C.); rita.galbiati@unimi.it (M.G.); valeria.crippa@unimi.it (V.C.); 2Jefferson Weinberg ALS Center, Vickie and Jack Farber Institute for Neuroscience, Sidney Kimmel Medical College, Department of Neuroscience, Thomas Jefferson University, Philadelphia, PA 19107, USA; 3Naxospharma srl, Novate Milanese, 20026 Milan, Italy; cristina.geroni@hotmail.it (C.G.); p.lombardi@naxospharma.eu (P.L.)

**Keywords:** misfolding, neurodegeneration, spinal and bulbar muscular atrophy, protein aggregation, berberine, amyotrophic lateral sclerosis, frontotemporal dementia, autophagy, proteasome

## Abstract

Background: Accumulation of misfolded proteins is a common hallmark of several neurodegenerative disorders (NDs) which results from a failure or an impairment of the protein quality control (PQC) system. The PQC system is composed by chaperones and the degradative systems (proteasome and autophagy). Mutant proteins that misfold are potentially neurotoxic, thus strategies aimed at preventing their aggregation or at enhancing their clearance are emerging as interesting therapeutic targets for NDs. Methods: We tested the natural alkaloid berberine (BBR) and some derivatives for their capability to enhance misfolded protein clearance in cell models of NDs, evaluating which degradative pathway mediates their action. Results: We found that both BBR and its semisynthetic derivatives promote degradation of mutant androgen receptor (ARpolyQ) causative of spinal and bulbar muscular atrophy, acting mainly via proteasome and preventing ARpolyQ aggregation. Overlapping effects were observed on other misfolded proteins causative of amyotrophic lateral sclerosis, frontotemporal-lobar degeneration or Huntington disease, but with selective and specific action against each different mutant protein. Conclusions: BBR and its analogues induce the clearance of misfolded proteins responsible for NDs, representing potential therapeutic tools to counteract these fatal disorders.

## 1. Introduction

Protein homeostasis (proteostasis) ensures proteome integrity and its proper regulation is essential for cell health and survival. The maintenance of proteostasis involves protein folding and degradation via a tight control mediated by the protein quality control (PQC) system. The PQC system requires the coordinated action of chaperones and degradative systems, namely the ubiquitin–proteasome system (UPS) and the autophagic-lysosomal pathway (ALP) [1,2]. UPS is involved in the fast protein degradation. Instead, ALP is responsible for clearing long-lived proteins, physically aggregated proteins and organelles [3].

When proteostasis is unbalanced, e.g., with ageing, the PQC system becomes unable to remove all the aberrant intracellular proteins efficiently, and some misfolded proteins might escape from intracellular defence. These events are worsened by genetic mutations, post-translation modifications or proteolytic cleavage that might confer misfolded conformations to the proteins, increasing their propensity to aggregate [4,5,6,7]. Protein misfolding and aggregation are common hallmarks of several neurodegenerative diseases (NDs), because neurons are postmitotic cells highly susceptible to the intracellular accumulation of damaged and misfolded proteins. Among these diseases, the most common are Parkinson’s disease (PD), Alzheimer’s disease (AD), the class of polyglutamine (polyQ) diseases, amyotrophic lateral sclerosis (ALS), frontotemporal lobar degeneration (FTLD), etc. [8]. Moreover, PQC system activity declines with aging and this might contribute to the adult-onset of NDs [9,10].

PolyQ diseases are a group of NDs caused by an abnormal polyQ expansion in causative and unrelated proteins. To date, nine polyQ diseases have been identified: spinal and bulbar muscular atrophy (SBMA), Huntington’s disease (HD), dentatorubral pallidoluysian atrophy (DRPLA), and six forms of spinocerebellar ataxia (SCAs) [11]. The polyQ size has a critical threshold specific for each disease that must be exceeded for disease onset. The size of the stretch is inversely correlated with the age of onset and symptoms severity [12]. SBMA is a rare X-linked neuromuscular disorder caused by polyQ expansion in the androgen receptor (AR), leading to a selective motoneuron degeneration occurring in the brainstem and spinal cord [13,14]. In SBMA only males are affected, and it has been estimated that there is a prevalence of 1:30,000 male inhabitants, even though the founder effect may account for geographical area variations [15,16]. SBMA is characterized by tremor, cramps and muscle weakness causing difficulties in walking, speaking or swallowing [17]. Signs of testicular atrophy, gynecomastia and reduced fertility may also be present [18,19]. All these clinical signs can be explained by an altered function of the mutant ARpolyQ in SBMA. In fact, the AR protein belongs to the steroid hormone nuclear receptor family and mediates the action of male androgens (testosterone (T) and dihydrotestosterone (DHT)) which regulate sexual male differentiation and other important biological functions such as the anabolic actions in skeletal muscle, and neuronal plasticity and neuroprotection in the central nervous system (CNS) [20,21,22]. ARpolyQ partially loses its transcriptional activity if compared to wild-type (wt) AR and this accounts for the endocrine signs identified in SBMA. However, complete androgen insensitivity syndrome, in which AR is completely inactive, does not result in neuronal loss suggesting that SBMA may be due to a gain of neurotoxic function of the mutant ARpolyQ. In fact, AR is highly expressed in the lower motoneurons and brainstem, as well as in the skeletal muscle, which are primary sites of SBMA pathogenesis [23]. The binding of androgens to AR normally induces conformational changes in the protein to allow its nuclear translocation, dimerization and interaction with specific DNA sequences [20]. In SBMA, the presence of the aberrantly long polyQ tract in AR may prevent its capability to reach the proper conformation after androgen activation, and the binding of androgens to ARpolyQ may induce its misfolding. It is important to note that the presence of the polyQ tract in AR protein is thought to induce structural alteration preventing its correct folding, possibly also altering the normal posttranslational modifications to which the wtAR undergoes right after its interaction with the ligand. Therefore, it is likely that the elongated polyQ in the mutant AR induces alteration on its intracellular trafficking and function. In addition, the exposure of the polyQ after ligand-induced Heat Shock Protein (HSP) dissociation causes protein misfolding which increases the propensity of the AR protein to aggregate. Misfolded ARpolyQ is poorly cleared from cells and may aggregate and perturb cells by inducing proteotoxic effects in affected motoneurons or muscle cells [24,25,26,27,28]. Thus, an enhanced ARpolyQ clearance should be beneficial against its neurotoxic properties.

The mechanism of protein misfolding found in SBMA has been also related to HD and other polyQ diseases, since they share the same type of mutation found in ARpolyQ even if present in other different proteins. HD is due to a polyQ expansion in the huntingtin (HTT) [29] a protein involved in embryogenesis and required for adult brain functions [30], being a scaffold protein implicated in a wide range of intracellular pathways, such as ALP, nuclear import, transcriptional regulation, apoptotic signaling and axonal transport [31,32,33].

In addition, SBMA also shares similarities with ALS, another motoneuron disease characterized by loss of upper and lower motoneurons causing progressive weakness and atrophy of skeletal muscles. ALS and SBMA involve the same cells types (with the exception of upper motor neurons spared in SBMA) and show similar symptoms [34,35,36]. However, only 10% of ALS cases are familial (fALS), while 90% of cases are sporadic (sALS). Some genes causing fALS, might also contribute to sALS [37]. Cu/Zn superoxide dismutase 1 (*SOD1*) gene mutations account for about 20% fALS, but misfolded wtSOD1 may also cause sALS [38,39]. TAR DNA-binding protein 43 (TDP-43) encoded by the *TARDBP* gene, is an RNA-binding protein found to accumulate into cytoplasmic inclusions in almost all sALS cases. The inclusions contain full length (FL) TDP-43, and some C-terminal fragments of TDP-43 (TDP-25 or TDP-35), generated by proteolytic cleavage by caspases or calpains [40]; the C-terminus of TDP-43 is poorly structured and highly prone to aggregate when released from the FL protein and by mislocalizing into the cytoplasm may seed for aggregate formation.

Despite the diverse aetiologies and the specific proteins involved, these NDs thus share a plethora of common characteristics and events that contribute to pathogenesis [41]; this may allow a possible common therapeutic approach aimed at stabilizing the native protein conformation, counteracting protein aggregation, or improving the misfolded protein clearance [34,35,42,43]. Several natural compounds, extracted from plants, are capable to regulate the PQC system and to exert protective effects in NDs [44]. An interesting compound is berberine (BBR), an isoquinoline alkaloid isolated from plants of *Berberidaceae* family, but also present in *Papaveraceae* and *Ranunculaceae* families. BBR is widely used in traditional Chinese medicine, and it has been shown to have a variety of pharmacological effects to attenuate inflammation, metabolic disorders, lipid metabolism, cardiovascular diseases, and it has been suggested to interfere with specific forms of cancer [45,46,47,48,49,50,51]. BBR has also been tested in *in vitro*, and in vivo models of AD, PD, HD, ALS/FTLD, where the compound was found to exert some neuroprotective effects [52,53,54,55,56,57]. Despite these findings, the mechanism of action of BBR remains poorly understood. Several intracellular mechanisms have been demonstrated to mediate the protective effects of BBR in neurons. The RAC-alpha serine/threonine-protein kinase (AKT) pathway and its downstream signaling such as apoptosis regulator BCL2 (BCL2), nuclear factor-kappaB (NF-kB), glycogen synthase kinase 3 beta (GSK3B), and mechanistic target of rapamycin kinase (MTOR) seem to be the principal mediators in promoting BBR effects against apoptosis, inflammation, oxidative stress and protein clearance in NDs [52,58,59,60,61,62]. BBR is able to cross the blood brain barrier (BBB), but it is poorly absorbed and rapidly metabolized [63,64,65,66,67,68]. Here, we studied the effects of BBR and BBR-derived NAX014, NAX035 and NAX117 (NAXs) on the aggregation and clearance of the ARpolyQ responsible for SBMA, and we also extended this evaluation on the clearance of misfolded proteins responsible for SBMA-related NDs. NAX014 presents a 4-chlorophenyl moiety and an ethylenic linker, whereas NAX035 and NAX117 a benzhydryl and a 4,4-dimethoxybenzhydryl moieties, respectively. NAXs were selected from a proprietary class of BBR semisynthetic structural analogues [U.S. Pat. 8188,109B2 to Naxospharma s.r.l.], characterized by the presence of (hetero)aromatic moieties, bonded to the 13-position of the parent alkaloid skeleton through a linker of variable length, in order to enhance the propensity for additional noncovalent aromatic interactions on the cellular target site(s) [69,70,71,72,73,74]. This could result in different biophysical effects with respect to the parent BBR and, hopefully, in a better biological outcome.

## 2. Results

### 2.1. Effects of BBR on ARpolyQ Clearance

To analyze the intracellular kinetics of wtAR and AR polyQ in NSC34 cell line, we used a chimeric protein coding for Green Fluorescent Protein (GFP) fused with wt AR (GFP-AR.Q22) or ARpolyQ (GFP-AR.Q48). These plasmid constructs were already tested and routinely used by us and other, and retain normal AR binding to androgens, nuclear translocation rate and proper transcriptional activity; in addition, the chimeric constructs showed the same biological behaviour and aggregation propensity of their related untagged plasmids [27,75,76,77]. In NSC34 cells, we observed that the wtAR is able to shuttle from cytoplasm to nucleus in response to the binding with the ligand testosterone. Otherwise, the mutant GFP-AR.Q48 is localized into the cytoplasm in its inactive status, but testosterone treatment induced the protein misfolding and the subsequent protein aggregation into the cytosol, confirming that SBMA is a ligand-dependent disorder (Figure 1A). In fact, ARpolyQ aggregation and neurotoxicity depend upon its testosterone activation, which also stabilizes the AR protein. This represents the situation found in patients, in which the high levels of testosterone in males activate ARpolyQ and trigger several intracellular pathways that contribute to disease pathogenesis. We initially assessed the specificity of the antibody both in Western Blot (WB) analysis and filter retardation assay (FRA) by using NSC34 cells expressing the exogenous recombinant AR with an elongated tract of 46 glutamines (AR.Q46) compared to a mock-transfected sample used as negative control. In microscopy analyses, to avoid cell permeabilization and immunoreaction with anti-AR antibody, which could give some unspecific autofluorescence signal in NSC34 cells, we selected a GFP-tagged mutant AR (GFP-AR.Q48) which also recapitulates the SBMA model without altering protein structure and function, since so far no differences have been reported between polyQ of such similar size (two Glns). Conversely, we demonstrated that both in WB and in FRA, the AR-H280 antibody was highly specific, since it detects only the band related to exogenous AR.Q46, while no signal is present in negative control samples, the untransfected NSC34 cells, which are known to be devoid of endogenous AR. Therefore, the selected antibody is highly specific for the AR proteins. In FRA experiments on the same samples, we confirmed that testosterone treatment enhanced AR.Q46 protein accumulation of high-molecular-weight species (Figure 1B). Moreover, in this case, no signals were detectable in cells devoid of endogenous AR, confirming that all immunoreactive materials retained on the cellulose acetate membrane have to be ascribed to aggregated forms of AR.Q46. To determine the optimal dose of BBR, the NSC34 cells were treated with BBR at a 0.2–10 μM concentration range for 48 h and cell viability was measured by 3-(4,5-dimethylthiazol-2-yl)-2,5-diphenyltetrazolium bromide (MTT) assay. The data indicate that BBR reduced cell viability starting from the dose of 0.5 μM, and this reduction was significant in a dose-dependent manner up to 10 µM BBR in comparison to control group. No significant difference was observed in cell viability at dose of 0.2 μM BBR, which resulted in being well tolerated by NSC34 cells (Figure 1C). Thus, we selected this dose of BBR to perform all further analyses in a cellular model of SBMA.

Treating NSC34 cells expressing AR.Q46 with BBR (used at three different nontoxic doses: 0.05–0.1–0.2 μM), we observed BBR induced the clearance of monomeric AR.Q46 species (Figure 1D), and, in parallel, reduced the accumulation of AR.Q46 insoluble species in a dose-dependent manner (this reduction was significant at BBR treatment of 0.1–0.2 μM (Figure 1E)). Of note, BBR also was able to enhance the clearance of the unactivated AR.Q46 evaluated in WB, which provides an estimation of the whole amount of Sodium dodecyl sulfate (SDS)-soluble protein in the samples; this suggests that even this form of the receptor, which is not fully folded yet, may undergo to a BBR-regulated enhanced degradation.

Since the effects of BBR could be exerted both/either at the level of protein translation and/or clearance, we used cycloheximide (CHX, a protein synthesis inhibitor) to test whether BBR acts on ARpolyQ synthesis and degradation. To this purpose, NSC34 cells expressing AR.Q46 were pretreated for one hour with CHX and then with BBR. In these conditions, we observed that BBR was still able to further reduce AR.Q46 levels in presence of CHX indicating that BBR prevents AR.Q46 accumulation and aggregation by promoting its degradation (Figure 2A,B for quantification).

To confirm this hypothesis, and to understand the main proteolytic pathway involved in BBR mechanism of action, we used inhibitors that selectively block the activity of UPS or ALP. We observed that the prodegradative activity of BBR on AR.Q46 was not affected by the autophagy inhibitor 3-methyladenine (3-MA). In fact BBR action on AR.Q46 clearance was not significant counteracted by ALP inhibition with 3-MA (Figure 2C,D). Conversely, we observed that BBR-mediated clearance of AR.Q46 was counteracted by the UPS selective inhibitor MG132, as shown by WB and FRA analysis (Figure 2E,F), suggesting that the BBR prodegradative activity on ARpolyQ is specifically mediated by the UPS activity.

### 2.2. Effects of BBR-Derived Compounds on ARpolyQ Clearance

To evaluate the potential toxicity of NAX014, NAX035 and NAX117 (the molecular structures of NAX compounds are illustrated in the Appendix A) on immortalized motoneurons, cell viability assays were initially performed using MTT analysis on NSC34 cells treated with NAXs at the same range of BBR doses, as described above. We observed that neither NAX014 nor NAX035 affected cell viability at a dose of 0.2 μM (Appendix A), while NAX117 was even better tolerated even using higher doses than the other two NAX compounds (Appendix A). Based on these data, we selected different concentrations of NAXs: 0.2 μM for NAX014 and NAX035, 0.5 μM for NAX117.

We tested the effects of NAXs on ARpolyQ clearance in comparison to BBR. As shown in the WB (Figure 3A) and FRA (Figure 3B) analyses, we observed that only NAX014 treatment significantly reduced AR.Q46 protein levels and aggregate accumulation, and this effect was similar to that exerted by BBR. NA0X35 and NAX117 induced only a slight and nonsignificant decrease in AR.Q46 accumulation (Figure 3A,B). Fluorescence microscopy analysis on cells transfected with the GFP-AR.Q48 and treated with testosterone together with BBR or NAXs revealed that all these compounds were able to reduce the percentage of cells bearing GFP-positive aggregates of GFP-AR.Q48, indicating that the three compounds are able to reduce or counteract ARpolyQ aggregation (Figure 3C and quantification in Figure 3D).

### 2.3. Effects of BBR and BBR-Derived Compounds on Misfolded Proteins Involved in ALS and HD

Since both BBR and the NAXs were found to be active on the mutant ARpolyQ activated by testosterone, we wondered whether these compounds were selective for this receptor or, instead, if they might exert a wider activity also against other misfolded proteins, thus possibly involved in a panregulation of the clearance of aggregating proteins in neurons. To this purpose, we also tested the activity of NAXs on three other mutant proteins or disease-associated protein fragments that are involved in ALS, FTLD and HD.

Firstly, we decided to test BBR and NAXs in cellular models of ALS and FTLD. To this end, we used NSC34 cells expressing genes coding for the Cu/Zn superoxide dismutase 1 (SOD1) with a missense mutation in position 93 (SOD1-G93A) as a fALS model, and for the C-terminal TDP-43 fragment of 25 KDa fused with the GFP (GFP-TDP-25) as a model for sALS and FTLD.

The data have shown that BBR only slightly reduces SOD1-G93A soluble and insoluble species, whereas all the NAXs robustly enhanced the clearance of mutant SOD1-G93A measured both in WB (SDS-soluble species) and in FRA (high-molecular-weight aggregated species of this misfolded protein). Interestingly, endogenous SOD1 levels were not affected by BBR or NAX treatment (Figure 4A,B).

By analyzing the BBR effects on the sALS model based on immortalized motoneurons, we found that this compound was able to induce GFP-TDP-25 clearance, both by reducing its immunoreactive protein in WB (SDS-soluble species) and the accumulation of high-molecular-weight aggregated species measured in FRA. These effects of BBR on GFP-TDP-25 also confirmed previous data reported by Chang et al. [54]. By analyzing the effects of the BBR-derived NAXs, we found that only NAX014 was able to significantly reduce the overall amounts of TDP-25 high-molecular-weight aggregated species measured in FRA, while both NAX014 and NAX117 reduced the levels of SDS-soluble GFP-TDP-25 in WB (Figure 4C,D).

Since the mutant ARpolyQ toxicity strictly depends upon the elongated polyQ tract, which is also the stretch responsible for neurotoxicity exerted by the mutant HTT in HD, we next wanted to find out whether BBR and the NAXs were also active in this different form of ND. In particular, the toxicity of the HTT protein containing the polyQ tract has been mainly associated with the release of a caspase-3 cleaved N-terminal fragment (coded by exon1) or with its generation via an incomplete splicing [78,79]. For this reason, NSC34 cells were transfected with a plasmid coding for the related N-terminus of the HTT protein (carrying 73 glutamines, HTT-exon1-73Q). Although its therapeutic effect was already demonstrated in an HD animal model [55], in our cell model, BBR did not show significant effects on the clearance of the mutant HTT, both at levels of the SDS-soluble species (evaluated in WB, Figure 4E) and of the high-molecular-weight aggregated species (evaluated in FRA, Figure 4F). Surprisingly, no effects of NAXs were observed on HTT-exon1-73Q insoluble species, except for the NAX035, which reduced the accumulation of high-molecular-weight aggregated species of HTT-exon1-73Q (evaluated in FRA) (Figure 4E,F).

## 3. Discussion

In this study we analyzed how BBR and some semisynthetic derivatives of BBR enhance the clearance of misfolded proteins associated to three different NDs, possibly providing a new method of intervention to counteract these diseases. Here, we assayed BBR activity in a cellular model of SBMA targeting the mutant ARpolyQ that exerts neurotoxic activities in motoneurons causing the disease. Our data suggest that BBR enhances the degradation of mutant and misfolded ARpolyQ via the UPS, while ALP blockage does not counteract ARpolyQ clearance induced by BBR. These data are in line with a previous report showing that BBR is able to suppress wtAR signaling via UPS activity in prostate cancer [80]. Despite the present work, the detailed mechanism of action of BBR on ARpolyQ has not been clarified, a protein-specific activity of BBR on AR might be involved, since, in prostate cancer, BBR was shown to disrupt the interaction between AR and HSP90, the chaperone involved in folding and degradation of AR [81], and protective in SBMA models [82,83,84,85,86]. These aspects may be particularly relevant for ARpolyQ, since it is known that HSP90, in cooperation with other chaperones, such as HSP70, maintains the AR (and its mutant ARpolyQ) in an inactive status confined into the cell cytoplasm, and only after ligand binding, HSPs are released and the AR migrates into the nucleus to exert its transcriptional regulation on androgen-dependent genes. In the case of ARpolyQ, the elongated polyQ tract may prevent the normal folding of the AR proteins, allowing its misfolding and aggregation. In fact, HSP release may expose a previously masked elongated polyQ causing its aggregation in the cytoplasm or in the nucleus. Several studies pointed out that the translocation and accumulation of ARpolyQ into the nucleus may be relevant for its toxicity [87,88,89,90,91,92]. Since BBR is able to reduce the overall accumulation of ARpolyQ in cells, this may also prevent the accumulation of misfolded ARpolyQ species inside the nucleus, thus preventing its nuclear toxicity. In addition to that, we demonstrated that BBR is also able to induce the degradation of other misfolded proteins, such as the 25 kDa C-terminal fragment of TDP-43, the TDP-25, involved in sALS (and in line with previous reports [54]), as well as mutant SOD1 causative of some forms of fALS. Of note, the effects of BBR on mutant SOD1-G93A may be associated to the antioxidant properties of BBR, which is able to maintain the activity of several antioxidant enzymes, including SOD1 [93], but it is likely that BBR also enhances the clearance of misfolded fraction of mutant SOD1-G93A prone to aggregate, thus exerting its neurotoxicity in affected neurons. In this context, the action of BBR seems to be rather specific, since we found that the levels of the endogenous SOD1 are not affected by BBR treatment, clearly indicating that BBR acts preferentially and specifically on misfolded protein clearance.

Since BBR was already tested in cellular and mouse models of HD, showing efficacy in promoting the clearance of mutant HTT by autophagic activation [55], we also evaluate if BBR was active in our cellular model of HD, but we did not confirm the data published, an effect possibly correlated to the different BBR dose used in our study compared to that previously reported [55] (5 to 100 μM) that were toxic in our cells.

Collectively, our data suggest that the neuroprotective properties of BBR in NDs might be primarily ascribed by its ability to potentiate intracellular degradative pathways, inducing misfolded protein clearance. However, how BBR enhances misfolded protein clearance remains obscure, since some reports point to an involvement of autophagy [54,55,60], while others to the UPS system [59,94,95,96]. In our case, we found that the enhanced degradation induced by BBR on ARpolyQ in SBMA is mainly associated to the UPS rather than to autophagy; additional studies are now in progress in our laboratory to fully characterize this mechanism from a molecular point of view.

Because of the positive effects of BBR in SBMA, we went further, analyzing three BBR semisynthetic derivatives, the NAXs compounds, which were designed to improve the activity of BBR in some tumors, such as colon, breast and pancreatic cancer. The NAXs are characterized by the presence of aromatic groups linked to the C-13 position, in order to allow additional non covalent interaction with the molecular targets [72,73,74,97]. In cancer models, these compounds resulted to be more effective than BBR, and represent promising anticancer drugs, while they have never been tested for their activity in NDs. Interestingly, we found that all NAXs reduced the number of cells bearing intracellular ARpolyQ aggregates, but only NAX014 was able to induce a significant clearance of the mutant protein, both in the soluble and in the insoluble forms. In addition, in cellular model of fALS expressing SOD1-G93A, all NAXs stimulated the degradation of mutant SOD1 and this effect was much more pronounced than that exerted by BBR. Conversely, in our cellular model of sALS (GFP-TDP25 expressing cells), only NAX014 exerted effects similar to BBR, while NAX035 was the unique compound effective in the clearance of mutant huntingtin in HD cell model. Moreover, these compounds seem to have a selective activity in response to specific misfolded proteins, for example NAX014 is effective on AR.Q46, TDP-25 and mutant SOD1, while it is ineffective or even detrimental on mHTT accumulation. This suggests that NAXs might activate different cell pathways with different effects in the presence of different type of misfolding proteins. Therefore, the chemical modifications to BBR skeleton might be able to induce selectivity on the mode of action of BBR, an interesting aspect, even if it remains to be determined whether these differences have to be ascribed to possible modification in their bioavailability or metabolism, as well as to their potential direct specific interaction with target misfolded proteins. Further studies aimed to better evaluate these mechanisms are in progress in our laboratory. At the same time, studies in animal models of NDs are required to determine if the compounds found active in cell models can be considered promising approach to ameliorate disease progression in NDs.

In this work, for the first time it has been demonstrated the beneficial effect of BBR in the clearance of ARpolyQ in motoneuronal cells. Although, a detailed mechanism of action has not been provided, BBR might be interesting for SBMA. While NAXs have been already tested in different types of cancer [72,73,74,97], this is the first report showing their potential use also in NDs. As mentioned above, some NAXs were found to have similar or more potent effects than BBR, but with different specific activities depending upon the misfolded protein target evaluated, suggesting that these compounds might be selective for mutant proteins. Given the pleiotropic and cell-specific effects exerted by BBR, and the different effects observed by NAX treatment on misfolded proteins clearance, future perspectives will be focused on unraveling the detailed mechanism of action of BBR on misfolded protein clearance, and, moreover, on understanding the different and specific activity of NAXs. It is possible that the different NAXs might activate different intracellular pathways leading to a selective clearance of misfolded proteins.

We believe that these data can be extended to other NDs, primarily caused by the formation and accumulation of misfolded proteins into specific neuronal populations. Indeed, compounds able to enhance misfolded protein clearance result in being very attractive for NDs, complex disorders generally characterized by the alteration of multiple intracellular functions and often by a slow progression. In this line, the use of natural compounds able to cross the BBB might represent a safe and useful treatment alone or in combination with other drugs.

## 4. Materials and Methods

### 4.1. Chemicals

Berberine chloride form (B3251), dimethyl sulfoxide (276855), testosterone (86500), Z-Leu-Leu-Leu-al/MG132 (C2211), 3-methyladenine (M9281), cycloheximide (C4859) were purchased from Sigma-Aldrich (Sigma-Aldrich, St. Louis, MO, USA). 13-[2-(4-chlorophenyl)ethyl]berberine iodide (NAX014), 13-(3,3-diphenylpropyl)berberine chloride (NAX035), and 13-[3,3-bis(4-methoxyphenyl)propyl]berberine chloride (NAX117) were provided by Naxospharma (Naxospharma s.r.l., Novate Milanese, Milan, Italy). MG132, 3-MA, BBR and NAXs were dissolved in dimethyl sulfoxide (DMSO), cycloheximide was dissolved in water, and testosterone was dissolved in ethanol.

### 4.2. Plasmids

EGFP-N1 plasmid was purchased from Clontech (Takara Bio Inc., Kusatsu, Japan). It was used in cotransfection with AR.Q46, pCDNA3-SOD1-G93A and HTT-exon1-73Q for controlling transfection efficiency. For AR biochemical analysis in the plasmid encoding for human AR with 46 polyQ size was used, kindly provided by Prof. M. Marcelli (Baylor College of Medicine, Houston, TX, USA [26]. For fluorescence microscopy analysis, the chimeric versions of AR tagged with GFP were used. GFP-AR.Q(n) plasmids encode a chimeric green fluorescent protein-AR with 22 or 48 polyQ stretch, which were kindly provided by Dr. Michael A. Mancini [98]. The pCDNA3-SOD1-G93A plasmid encoding G93A mutant SOD1 was kindly provided by Dr. C. Bendotti (Mario Negri Institute for Pharmacological Research, Milan, Italy). The GFP-TDP-25 encodes GFP protein fused with the 25-kDa C-terminal fragment of TARDBP was kindly provided by Prof. L. Petrucelli (Mayo Clinic, Jacksonville, FL, USA). The HTT-exon1-73Q plasmid contains the cDNA corresponding to exon-1 of *HTT* encoding the mutant HTT N-terminus (exon-1) with a polyQ of 73 glutamines and it was kindly provided by Dr. M. Basso (University of Trento, Trento, Italy).

### 4.3. Cell Cultures and Transfections

The mouse motor-neuron-like hybrid cell line (NSC34), kindly provided by Dr. N.L. Cashman (University of British Columbia, Vancouver, CAN) [99], was maintained in DMEM medium supplemented with 5% fetal bovine serum (FBS)(F7524, Sigma-Aldrich), 1 mM L-glutamine (ECB3004D, Euroclone, Pero, Italy), and penicillin-streptomycin (penicillin, 31749.04, SERVA Electrophoresis GmbH, Heidelberg, Germany; streptomycin, 35500.01, SERVA Electrophoresis GmbH), at 37 °C in 5% CO_2_. For experiments involving testosterone treatment, a dextran-treated charcoal-stripped FBS was used to selectively remove hormones [100,101]. For the expression of the exogenous mutant proteins, NSC34 cells were transiently transfected with 0.7 μg of plasmid DNA using transferrin (T8150, Sigma-Aldrich) in conjunction with Lipofectamine reagent (18324012, Thermo Scientific Life Sciences Research, Waltham, MA, USA) to improve transfection efficiency.

### 4.4. MTT Assay

MTT (3-(4,5-dimethylthiazolyl-2)-2,5-diphenyltetrazolium bromide) (M5655, Sigma-Aldrich) assay was performed to test cellular metabolic activity, and cell viability. NSC34 were seeded at 45,000 cell/well in 24-well plates, treated with different doses of BBR or NAXs as described in Figure legends. After 48 h of treatment, the medium was removed, cells were incubated for 30 min with 300 μL of MTT at 37 °C, then 500 μL of 2-propanol were added to each well as dissolving solvent. Absorbance was measured at OD = 570 nm with an Enspire^®^ Multimode plate reader (PerkinElmer, Waltham, MA, USA). The experiments were performed with six independent samples (n = 6).

### 4.5. Western Blot and Filter Retardation Assay

For WB and FRA, cells were seeded at 90,000 cells/well in a 12-well plate, and transfected/treated as described in the text. Forty-eight hours after transfection, the cells were collected and centrifuged at 500 g for 5 min at 4 °C. The pellets were lysed in PBS solution supplemented with protease inhibitors (Sigma-Aldrich, P8340) using slight sonication. Protein extracts were quantified with the bicinchoninic acid method (EMP014500, Euroclone, EMP014500, Pero, Italy). For WB analysis, 20 μg of proteins were loaded on 12% SDS-polyacrylamide gel electrophoresis and transferred to nitrocellulose membrane by the Trans-Blot^®^ Turbo™ transfer system (Bio-Rad Laboratories, 1704150, (Bio-Rad, Hercules, CA, USA). FRA is a technique routinely used in our laboratory to measure the high-molecular-weight species or insoluble proteins. Sample filtration is based on the membrane composition (cellulose acetate membrane does not have affinity for proteins) and porosity. Only insoluble proteins remain entrapped on the membrane [35,36,43,102]. For FRA, 3 μg of proteins were loaded and filtered on cellulose acetate in a slot-blot apparatus (1703938, Bio-Rad Laboratories, 1703938). FRA membranes were processed following the WB protocol. The membranes were incubated for 1 h in blocking solution (1X-TBST with 5% *w/v* nonfat dried milk), followed by overnight incubation with diluted primary antibody in blocking solution. The following primary antibodies were used: anti-AR (H280; Santa Cruz Biotechnology, Dallas, TX, USA), (sc-13062; 1:1000), anti-GFP for TDP (Abcam, Cambridge, UK) (AB1218; 1:2000), anti SOD1 (Enzo Life Sciences, Farmingdale, NY, USA) (ADI-SOD-100, 1:1000), anti-HTT (Sigma-Aldrich), (MAB2166, 1:2000), anti-GAPDH (FL-335; Santa-Cruz Biotechnology), (sc-25778; 1:3000). After incubation, the blots were washed and incubated for 1 h with peroxidase-conjugated secondary antibodies (Jackson Immunoresearch Laboratories, West Grove, PA, USA) (111-035-003 goat antirabbit; 115-035-003 goat antimouse; 1:5000). Chemiluminescent substrate detection was performed using Clarity^TM^ ECL Western blotting substrate (1705060, Bio-Rad Laboratories, 1705060) and Chemidoc XRS System (1708265, Bio-Rad). Quantification analysis was performed using Image Lab Software, version 5.2.1 (Bio-Rad). For WB data quantification and normalization, the band intensity of the target protein was divided by the intensity of the loading control protein, GAPDH. Three independent biological samples for each condition were analyzed and quantified. Then, we performed a comparative analysis of relative target protein levels across sample expressed as fold difference compared to the mean value of lane 1 intensity (normalized density of each line divided by the mean value of normalized density of lane 1 samples).

### 4.6. Fluorescence Microscopy Analysis

For fluorescence microscopy analysis, we used the chimeric version of AR, the GFP-ARQn plasmids. The cells were seeded at 35,000 cells/well in 24-well plates on 13-mm coverslips, transfected and treated as described in figure legends. The cells were fixed with 4% paraformaldehyde in PBS and the nuclei were stained with DAPI (1:10,000 in PBS). Coverslips were mounted onto slides with Mowiol^®^ 4-88 (475904, Merck-Millipore, Burlington, MA, USA) and analyzed with an Axiovert 200 microscope (Zeiss, Oberkochen, Germany) equipped with a Photometric Cool-Snap CCD camera (Ropper Scientific, Trenton, NJ, USA). The percentage of cells with aggregates was calculated by dividing the number of cells with aggregates by the total number of transfected cells and was performed by manual counting of three fields per sample (n = 3), using PL10X/20 eyepiece with graticules. The fields were randomly selected by a blinded observer. Representative images of fluorescence microscopy analysis were acquired using Zeiss LSM900 laser scan microscope and analyzed with Zen software (Zeiss).

### 4.7. Statistical Analysis

The data are presented as mean ± standard deviation. Statistical analysis has been performed using PRISM (version 7) software (GraphPad Software, San Diego, CA, USA). Analysis of variance (ANOVA) was done to compare three or more groups: one-way ANOVA for single independent variable, two-way ANOVA for two independent variables. *p* value <0.05 was considered statistically significant. A post hoc test was performed (see figure legends for details).

## Figures and Tables

**Figure 1 ijms-21-03443-f001:**
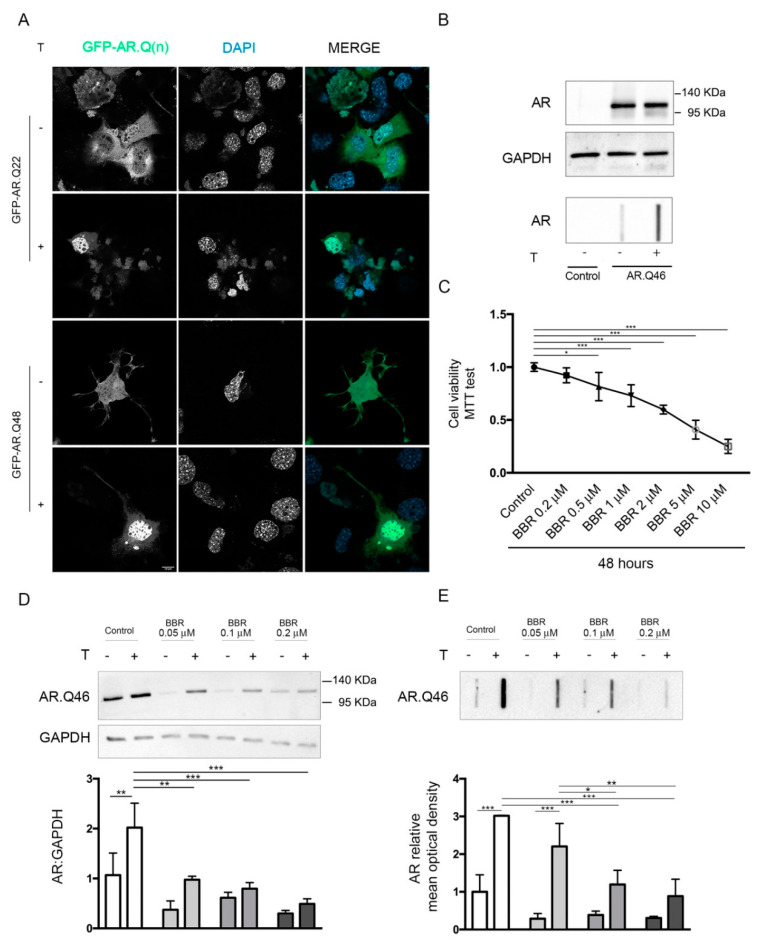
BBR reduced the ARpolyQ protein level. (**A**) Representative confocal microscopy analysis of GFP-AR.Q(n) intracellular localization. NSC34 cells were transfected with GFP-ARQ.22 or GFP-AR.Q48 in absence or in presence of 10 nM testosterone. Nuclei were stained with DAPI (63X magnification). Scale bar: 10 μM. (**B**) WB analysis performed on NSC34 cells transfected with AR.Q46 in absence or in presence of 10 nM testosterone, untransfected control was added to test the antibody specificity. GAPDH was used as loading control. (**C**) MTT cell viability assay was performed on NSC34 cells treated with BBR at different concentration for 48 h (* *p* < 0.05, *** *p* < 0.001, one-way ANOVA, followed by Tukey’s test). (**D**,**E**) NSC34 cells were transfected with AR.Q46 in absence or presence of 10 nM testosterone and BBR at three different doses (0.05, 0.1, and 0.2 μM for 48 h Ethanol and DMSO were used as vehicle control for testosterone and BBR, respectively. (**D**) WB analysis was performed. GAPDH was used as loading control, and the bar graph represents the mean optical density ± SD of AR: GAPDH (n = 3) (**E**) FRA was performed, the bar graph represents the mean optical density of AR ± SD (n = 3). (* *p* < 0.05, ** *p* < 0.01, *** *p* < 0.001, two-way ANOVA, followed by Tukey’s test).

**Figure 2 ijms-21-03443-f002:**
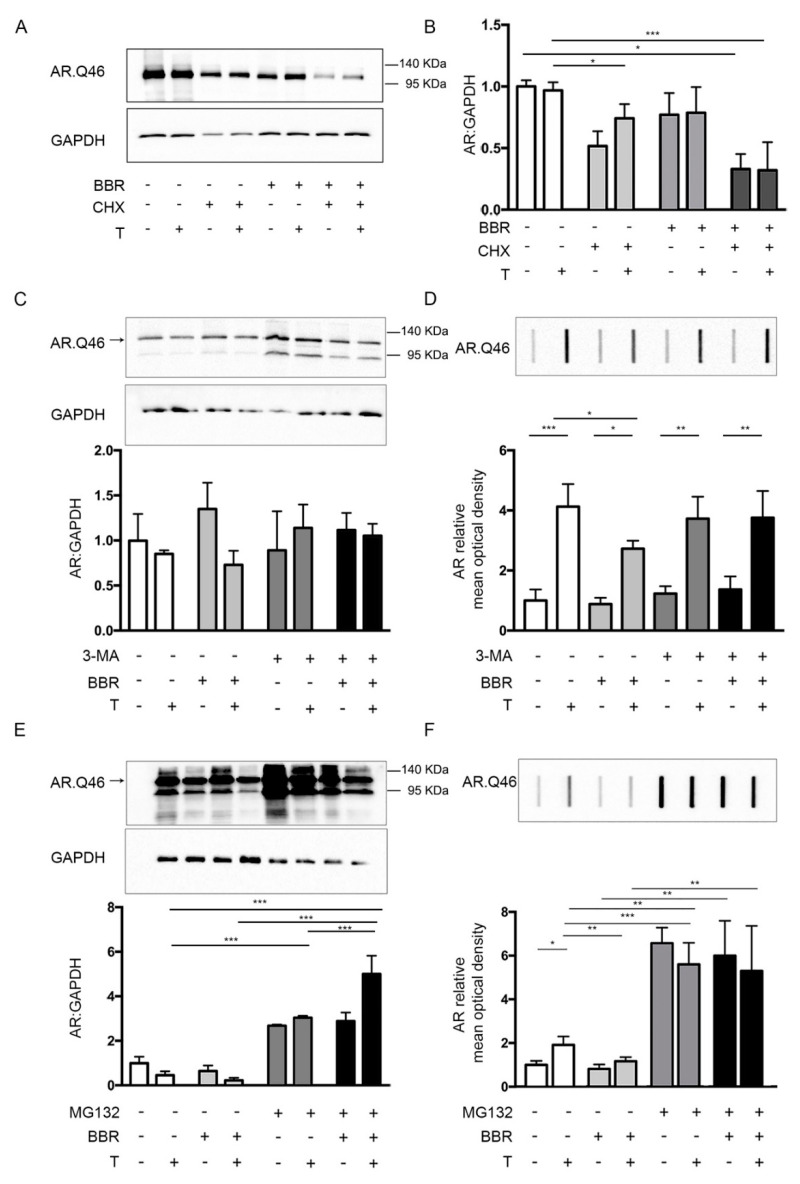
BBR pro-degradative activity on ARpolyQ. (**A**) WB analysis on NSC34 cells transfected with AR.Q46 in absence or in presence of 10 nM testosterone. To inhibit protein synthesis the cells were pretreated with 20 μΜ cycloheximide (CHX) and then treated with 0.2 μΜ BBR or DMSO as vehicle control for 48 h. GAPDH was used as loading control. (**B**) The bar graph represents the mean optical density ± SD of AR: GAPDH (n = 3) (* *p* < 0.05, *** *p* < 0.001, two-way ANOVA, followed by Tukey’s test). (**C**,**F**) NSC34 cells transfected with AR.Q46 in absence or in presence of 10 nM testosterone and treated with 0.2 μΜ BBR and with 10 mM 3-MA (**C**,**D**) or 10 μΜ MG132 (**E**,**F**) to inhibit autophagic or proteasomal activity, respectively. (**C**,**E**) WB analyses were performed; GAPDH was used as loading control; the bar graphs represent the mean optical density ± SD of AR: GAPDH (n = 3). *** *p* < 0.001, two-way ANOVA, followed by Tukey’s test). (**D**,**F**) FRA was performed. The bar graphs represent the mean optical density of AR ± SD (n = 3). (* *p* < 0.05, ** *p* < 0.01, *** *p* < 0.001, two-way ANOVA, followed by Tukey’s test).

**Figure 3 ijms-21-03443-f003:**
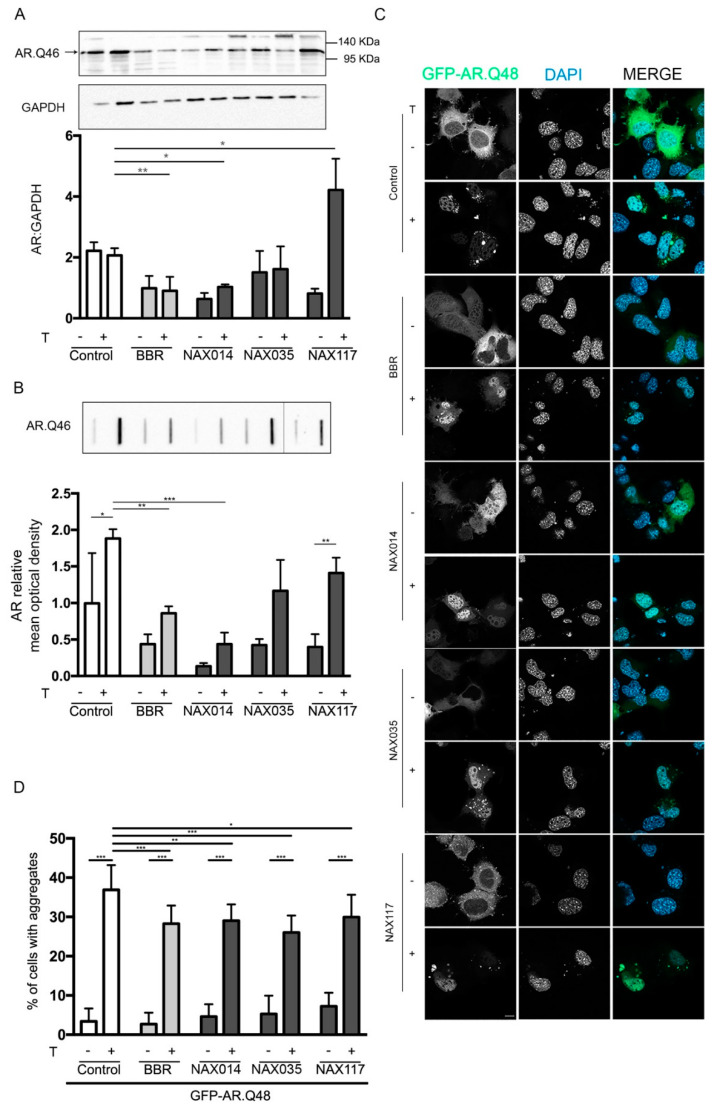
Effects of NAXs on ARpolyQ. (**A**,**B**) NSC34 cells were transfected with AR.Q46 in absence or in presence of 10 nM of testosterone and treated with BBR or NAX compounds for 48 h (BBR, NAX014 and NAX035 were used at 0.2 μM, NAX117 was used at 0.5 μM). (**A**) WB analysis was performed, GAPDH was used as loading control, the bar graph represents the mean optical density ± SD of AR: GAPDH (n = 3) (* *p* < 0.05, ** *p* < 0.01, two-way ANOVA, followed by Tukey’s test). (**B**) FRA was performed. The bar graph represents the mean optical density of AR ± SD (n = 3) (* *p* < 0.05, ** *p* < 0.01, *** *p* < 0.001, two-way ANOVA, followed by Tukey’s test). (**C**,**D**) Fluorescence microscope analysis performed on NSC34 cells transfected with GFP-ARQ48 in absence or in presence of 10 nM testosterone and treated BBR or NAX compounds for 48 h (BBR, NAX014 and NAX035 were used at 0.2 μM, NAX117 was used at 0.5 μM). (**C**) Representative fluorescence microscopy analysis of GFP-AR.Q48 protein levels. Nuclei were stained with DAPI (63X magnification). Scale bar: 10 μM. (**D**) The quantification of GFP-AR.Q48 aggregates was performed using a PL 10X/20 eyepiece with graticules. The percentage of cells with aggregates was obtained dividing the number of cells with aggregates with the total number of GFP positive cells (* *p* < 0.05, ** *p* < 0.01, *** *p* < 0.001, two-way ANOVA, followed by Tukey’s test).

**Figure 4 ijms-21-03443-f004:**
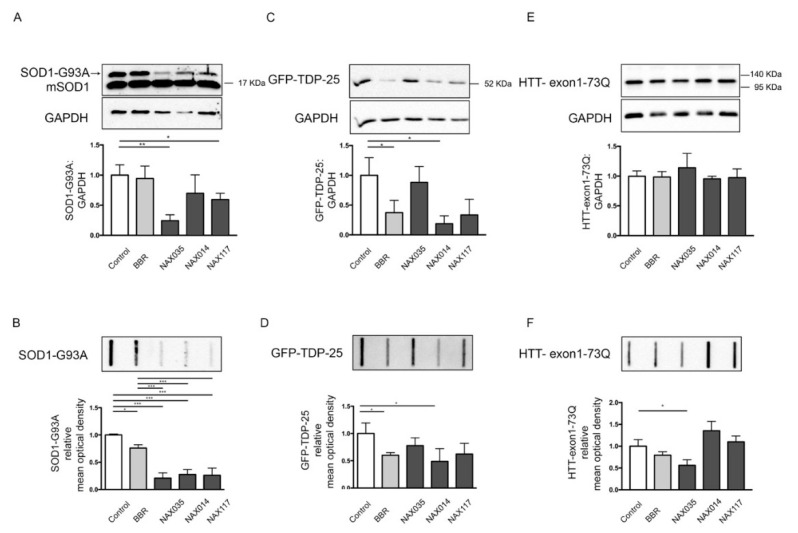
Effects of NAXs on misfolded proteins involved in ALS and HD. (**A**,**B**) NSC34 cells were transfected with plasmids coding for SOD1-G93A and treated BBR or NAX compounds for 48 h (BBR, NAX014 and NAX035 were used at 0.2 μM, NAX117 was used at 0.5 μM). (**A**) WB analysis was performed with anti-SOD1 antibody to detect exogenous human SO1-G93A and the endogenous murine SOD1 (mSOD1). GAPDH was used as loading control, the bar graph represents the mean optical density ± SD of SOD1-G93A: GAPDH (n = 3) (* *p* < 0.05, ** *p* < 0.01, one-way ANOVA, followed by Tukey’s test). (**B**) FRA was performed, the bar graph represents the mean optical density of SOD1-G93A ± SD (n = 3) (* *p* < 0.05, *** *p* < 0.001, one-way ANOVA, followed by Tukey’s test). (**C**,**D**) NSC34 cells were transfected with plasmids coding for GFP-TDP-25 and treated with BBR or NAX compounds for 48 h (BBR, NAX014 and NAX035 were used at 0.2 μM, NAX117 was used at 0.5 μM). (**C**) WB analysis was performed with anti-GFP antibody to detect the GFP-TDP-25. GAPDH was used as loading control, the bar graph represents the mean optical density ± SD of GFP-TDP-25: GAPDH (n = 3) (* *p* < 0.05, one-way ANOVA, followed by Tukey’s test). (**D**) FRA was performed, the bar graph represents the mean optical density of GFP-TDP-25 ± SD (n = 3) (* *p* < 0.05, followed by Tukey’s test). (**E**,**F**) NSC34 cells were transfected with plasmids coding for HTT-exon1-73Q and treated BBR or NAX compounds for 48 h (BBR, NAX014 and NAX035 were used at 0.2 μM, NAX117 was used at 0.5 μM). (**E**) WB analysis was performed with anti-HTT antibody to detect exogenous HTT-exon1-73Q. GAPDH was used as loading control, the bar graph represents the mean optical density ± SD of HTT-exon1-73Q: GAPDH (n = 3) (one-way ANOVA, followed by Tukey’s test). (**B**) FRA was performed, the bar graph represents the mean optical density of HTT-exon1-73Q ± SD (n = 3) (* *p* < 0.05, one-way ANOVA, followed by Tukey’s test).

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
