# Peer review of "Enhanced Clearance of Neurotoxic Misfolded Proteins by the Natural Compound Berberine and Its Derivatives"

_ijms, 2020, doi:10.3390/ijms21103443_

Round 1
Reviewer 1 Report
Article title “Enhanced Clearance of Neurotoxic Misfolded Proteins by the Natural Compound Berberine and Its Derivatives” by Rusmini et al is well written, carefully designed and conducted experiments, logical interpretation of obtained data and their correlation with the previous finding. This paper improves our current understanding of the natural compound effect on several aggregations associated diseases. The effect of berberine on the induction of pathway to clear off several aggregate species in the cell appears to be convincing and supported by several lines of experimental evidence but this paper still fails to come up with any mechanistic details of mode of action of this natural compound. My suggestion would be to come up with some possible models for the BBR mode of action. Some of the specific suggestions and comments are mentioned below.
- I will suggest to include a couple of lines about the differences between AR-wt and mutant AR in the introduction.
- The author should explain why did they use two different forms of ARpolyQ (GFP-ARpolyQ48 for microscopy and AR.Q46 for WB/FRA.
- As mentioned in line no 141, ARpolyQ aggregation requires activation by testosterone. Does this aggregation a concentration-dependent phenomenon?
- Please check figure 1 labeling.
- All WB are missing molecular weight markers. It is advisable to include them because, in figure 2E, there are multiple bands, and its not clear which band is used for quantification.
- Explain the mean optical density calculation in the method section for clarity.
- To handle the increased level of misfolded protein in the cell, there would be an increased activation of the proteasome system. I will suggest the author to check whether BBR is modulating the regulation of UPS gene or BBR directly acting on UPS pathway protein(s) and over-activate them to degrade ARpolyQ protein?
- Figure 3 can be moved to supplement.
- Again, figure 5A, information missing about which band has been used for quantification. Use MWt marker.
- In the introduction section, the author mentioned that presence of testosterone induce misfolding followed by the aggregation of mutant ARpolyQ protein, and in the discussion section, they discussed that BBR acts on misfolded protein and leads to their degradation through UPS. If this is the case then why is the protein level so low in BBR treated samples (Fig 1D) even in the absence of testosterone where protein is expected to be in folded form. If we compare the Fig1E (FAR experiment) lane with no T/+BBR, there is no protein band suggesting that ARpolyQ indeed in the soluble form. They should explain this.
Author Response
We thank Reviewer n. 1 for the time and skill He/She devoted to our manuscript and for his/her very positive comments on our data that BBR and derivatives induce the clearance of several aggregate species. We are aware that our data still lack some more deeper mechanistic details of the mode of action of these compounds. At present we are able to prove that the proteasomal pathway mediates the pro-degradative effects of BBR, while we excluded that autophagy plays a major role in this context, which is already a frame from which we would like to start a new project. In fact, we were working deeply on these aspects, but unfortunately our lab is under lockdown, since two months and we have no idea on when we will be able to return full time to our research activities. We thus have planned to perform studies aimed to unravel more deeply the mode of action of these compounds for a new manuscript entirely dedicated to these aspects, that we would like to prepare in the (we hope, near) future. Meanwhile, it is our aim that sharing these data with the scientific community involved in protein misfolding disease studies may also stimulate other researchers to look for the mode of action of BBR, and this may speed up the accumulation of knowledge on these compounds as well as their possible therapeutic potential.
With regards to the specific comments raised, we are pleased to answer as follow:
Point 1 - I will suggest to include a couple of lines about the differences between AR-wt and mutant AR in the introduction.
Reply to Point 1. We are sorry, we actually did not mention this aspect in our introduction. Thanks for noting it. We added a paragraph in the introduction with a description of the role of the polyQ in the AR mutant protein and explained the differences between AR-wt and polyQ AR.
Point 2 - The author should explain why did they use two different forms of ARpolyQ (GFP-ARpolyQ48 for microscopy and AR.Q46 for WB/FRA.
Reply to Point 2. In reality this is a very simple technical aspect which allow us to going through cell permeabilization and immunoreaction in fluorescence microscopy analysis, avoiding the use of commercially available antibodies against AR (which we found not excellent for IF in NSC34 cells, but very good for WB and FRA as shown in the results). For WB and FRA we prefer to use the untagged protein as we did in many of our publication in the past. The construct with 48 and 46 Gln are almost identical, since variation of two glutamines in the polyQ size is not expected to cause a different biochemical behavior. We just obtained a little expansion during cloning of the GFP-tagged vectors (performed in the lab of Prof Mancini). We added this explanation at the very beginning of the Result section.
Point 3 - As mentioned in line no 141, ARpolyQ aggregation requires activation by testosterone. Does this aggregation a concentration-dependent phenomenon?
Reply to Point 3. We have not performed a kinetic analysis of the rate of aggregation of ARpolyQ in function of the ligand dose. We selected a concentration of testosterone (10 nM), which is generally used in literature (by other and us), since it is able to fully saturate the AR binding site even when the protein is overexpressed. The selected dose is approximately 5-10 time higher that the KD of this ligand for its receptor (KD has been estimated to vary between 1 and 2 nM for Testosterone).
Point 4 - Please check figure 1 labeling.
Reply to Point 4. Thanks for noting this. We apologize for the mistake, we have corrected the labels in the figure as required.
Point 5 - All WB are missing molecular weight markers. It is advisable to include them because, in figure 2E, there are multiple bands, and its not clear which band is used for quantification.
Reply to Point 5. Again, thanks for noting this. We apologize for this inaccuracy. In the revised version of the manuscript we included the molecular weight of marker used in the western blots, and we have indicated the bands used for quantification with arrows as suggested.
Point 6 - Explain the mean optical density calculation in the method section for clarity.
Reply to Point 6. We included an explanation of the density calculation in the method section as requested.
Point 7. To handle the increased level of misfolded protein in the cell, there would be an increased activation of the proteasome system. I will suggest the author to check whether BBR is modulating the regulation of UPS gene or BBR directly acting on UPS pathway protein(s) and over-activate them to degrade ARpolyQ protein?
Reply to Point 7. We thank the Reviewer for suggesting us this experiment. As mentioned above, at present we have no possibility to perform this analysis, because our laboratory is closed and we do not know when we will be able to return to normal activities in the future. We plan to complete a study aimed to unravel the mechanism by which BBR stimulate or facilitate misfolded protein clearance which will be incorporate in a future manuscript to be prepared once the data will become available. As you may understand, we are very sorry we cannot provide better answer to this question.
Point 8. Figure 3 can be moved to supplement.
Reply to Point 8. We agree with this suggestion. The data are not essential to the general flow of the study. We have moved the figure to the supplement, and now named Figure S1.
Point 9. Again, figure 5A, information missing about which band has been used for quantification. Use MWt marker.
Reply to Point 9. Sorry also for this inaccuracy, and many thanks for noting it. We have included the molecular weight of marker used in the western blots, and we have indicated the bands used for quantification with arrows.
Point 10. In the introduction section, the author mentioned that presence of testosterone induce misfolding followed by the aggregation of mutant ARpolyQ protein, and in the discussion section, they discussed that BBR acts on misfolded protein and leads to their degradation through UPS. If this is the case then why is the protein level so low in BBR treated samples (Fig 1D) even in the absence of testosterone where protein is expected to be in folded form. If we compare the Fig1E (FAR experiment) lane with no T/+BBR, there is no protein band suggesting that ARpolyQ indeed in the soluble form. They should explain this.
Reply to Point 10. This is an interesting point. With regard to the insoluble fraction detected on FRA, we should mention that, since in basal condition the unactivated (not treated with testosterone) ARpolyQ does not aggregate and the amount found on the cellulose acetate membrane, in which we measure the insoluble material is at the background levels; in fact, there is no effect of BBR on this parameter. In WB, in which we measure the overall levels of the SDS-soluble ARpolyQ, there is a mild effect of BBR. We already published that also the exogenous ARpolyQ when not activated by testosterone is degraded by the proteasome (10.1016/j.nbd.2010.08.023); in fact, unbound ARpolyQ is not misfolded, but (with the exception to the ligand binding pocket) is not fully folded, yet, since the final conformation is obtained only after Testosterone binding and AR dissociation from accessory proteins. Thus, even the still unfolded ARpolyQ may undergoes to normal turnover in the cells. It is likely that BBR, which we hypothesized stimulates the proteasomal pathway, may also influence the clearance of the unactivated ARpolyQ. Since this is an interest point, that we did not note while preparing the manuscript, we have added a sentence in which we mention this possibility in the Result section related to this figure. Thanks for noting this phenomenon.
Reviewer 2 Report
In the manuscript submitted to IJMS (804268) authors works on the enhanced clearance of neurotoxic mMisfolded proteins by the natural compound berberine and Its derivatives. This reviewer suggest the publication in IJMS after minor revision.
Title and Abstract are adquated.
Introduction is well constructued and Literature adequated and recent.
Instrumental techniques, as the use of plasmids, cell cultures and transfections, MTT assay and Western Blot, filter retardation assay,and fluorescence microscopy analysis.
Statistical analysis is very useful for interpretation of the data
Results described in ‘Effects of BBR on ARpolyQ clearance’, ‘Effects of BBR-derived compounds on ARpolyQ clearance’, and ‘Effects of BBR and BBR-derived compounds on misfolded proteins involved in ALS and HD’, are interesting. Also the Disccussion obtained about them.
Figures are adequated in number and quality.
As a minor comments, in Experimental, for Instrumentation, Materials and Reagents, or Programs and Databases (as SPSS, Excel, and others) ever, Product (Manufacturer, City, Country), in this order and format. Please correct in some places. In the case of USA products: Product (Manufacturer, City, State, USA).
Author Response
We thank Reviewer n. 2 for the very positive comments provided on our manuscript.
With regards to the minor comment:
Point 1. in Experimental, for Instrumentation, Materials and Reagents, or Programs and Databases (as SPSS, Excel, and others) ever, Product (Manufacturer, City, Country), in this order and format. Please correct in some places. In the case of USA products: Product (Manufacturer, City, State, USA).
Reply to point 1. We are pleased to say that we have corrected the details about the products described in the methods, as requested.